# The Potential Impact of Smog Spell on Humans’ Health Amid COVID-19 Rages

**DOI:** 10.3390/ijerph182111408

**Published:** 2021-10-29

**Authors:** Ammar Javed, Farheen Aamir, Umar Farooq Gohar, Hamid Mukhtar, Muhammad Zia-UI-Haq, Modhi O. Alotaibi, May Nasser Bin-Jumah, Romina Alina Marc (Vlaic), Oana Lelia Pop

**Affiliations:** 1Institute of Industrial Biotechnology, Government College University Lahore, Lahore 54000, Pakistan; ammarjaved94@gmail.com (A.J.); farheenaamir15@gmail.com (F.A.); dr.mufgohar@gcu.edu.pk (U.F.G.); hamidmukhtar@gcu.edu.pk (H.M.); 2Office of Research, Innovation & Commercialization, Lahore College for Women University, Lahore 54000, Pakistan; 3Biology Department, College of Science, Princess Nourah Bint Abdulrahman University, Riyadh 11671, Saudi Arabia; mouotaebe@pnu.edu.sa (M.O.A.); mnbinjumah@pnu.edu.sa (M.N.B.-J.); 4Environment and Biomaterial Unit, Health Sciences Research Center, Princess Nourah Bint Abdulrahman University, Riyadh 11671, Saudi Arabia; 5Food Engineering Department, Faculty of Food Science and Technology, University of Agricultural Sciences and Veterinary Medicine, 400372 Cluj-Napoca, Romania; 6Department of Food Science, University of Agricultural Science and Veterinary Medicine, 400372 Cluj-Napoca, Romania; oana.pop@usamvcluj.ro

**Keywords:** air pollution, COVID-19, photochemical smog, respiratory disorders

## Abstract

Rapid and unchecked industrialization and the combustion of fossil fuels have engendered a state of fear in urban settlements. Smog is a visible form of air pollution that arises due to the over-emissions of some primary pollutants like volatile organic compounds (VOCs), hydrocarbons, SO_2_, NO, and NO_2_ which further react in the atmosphere and give rise to toxic and carcinogenic secondary smog components. Smog reduces the visibility on roads and results in road accidents and cancellation of flights. Uptake of primary and secondary pollutants of smog is responsible for several deleterious diseases of which respiratory disorders, cardiovascular dysfunction, neurological disorders, and cancer are discussed here. Children and pregnant women are more prone to the hazards of smog. The worsening menace of smog on one hand and occurrence of pandemic i.e., COVID-19 on the other may increase the mortality rate. But the implementation of lockdown during pandemics has favored the atmosphere in some ways, which will be highlighted in the article. On the whole, the focus of this article will be on the dubious relationship between smog and coronavirus.

## 1. Introduction

The word smog is an amalgamation of two words, ‘smoke’ and ‘fog’. Fog is reckoned as a visible low lying cloud, made up of small water droplets or ice crystals [1]. In 1905, H. A. Des Voeux used the term ‘smog’ to define the atmospheric conditions of many towns in Britain. In 1911, this word became famous when H. A. Des Voeux reported 1000 deaths in his paper ‘Smoke and Fog’ due to ‘smoke-fog’ in Edinburgh and Glasgow [2]. Nowadays, Lahore, Faisalabad, Delhi, Beijing, Los Angles, Mexico, and London are mostly affected by the smog [3,4,5,6,7,8,9].

Currently, various studies have provided knowledge to the general population about the relationship between smog and its adverse effects on human health. Earlier researchers had confirmed that health effects are related to persons’ age, health, and socioeconomic status [10]. However, the impact of smog is also influenced by its time of exposure. The risk of long-term exposure is much higher than that of short-term exposure. Both the long-term unceasing exposure and short-term peak do not have the same consequences and they follow different dynamics. The effects range from short-term irritation in the trachea to long-term genetic mutations. However, some recent studies have observed a link between adverse mortality and short-term exposure to smog [11]. These adverse health effects have a broad array from subclinical effects like irritation in the trachea to long-term genetic mutations and premature deaths. Some of the major diseases which are harbored by smog are respiratory diseases (asthma, coughing, and bronchiolitis), cardiovascular disease, neurological disorders, cancer, infant health, low birth weight, and other problems like eye irritation and breathing difficulties [12,13,14,15].

On the other hand, the novel coronavirus outbreak has shaken the world. It engulfed the whole world within a year. The root cause of this pandemic i.e., SARS-CoV-2 (Severe acute respiratory syndrome coronavirus-2) is transmissible from humans to humans. It targets the respiratory tract of humans, attaches with the angiotensin-converting enzyme-2 (ACE2), and down-regulates its production to cause severe respiratory illnesses. Although the mortality rate of coronavirus disease of 2019 (COVID-19) is less than 10% together with smog, the increase in fatality can be observed because both target the respiratory tract of humans [16]. In elderly people, coronavirus invasion is facilitated as immune responses are weakened by age and smog [17]. Together they may worsen the disease and can lead to hospitalization and eventually death occurs [18] as shown in Figure 1. Studies have backed the hypothesis that components of air pollution like nitrogen dioxide (NO_2_) and particulate matter (PM) cause the excessive production of the Angiotensin-converting enzyme-2 (ACE2) which is the binding target for SARS-CoV-2 [19,20,21]. This increased production increases the susceptibility towards COVID-19. Thus the combination of coronavirus and air pollution can exacerbate the situation. The regions like China, India, and the USA a positive correlation is observed between COVID-19 mortality and high air pollution. This aggravates the need to control air pollution to reduce coronavirus cases where they share common hotspots [22,23]. However, there is another aspect of this pandemic. Due to the pandemic, people are locked in their houses to avoid SARS-CoV-2 infection. This reduction in human activities has a positive has brought a positive impact on nature like less water, air, and noise pollution. The lockdown periods have also prevented several deaths due to a reduction in air pollutants [24]. The purpose of the paper is to highlight the negative aspects of the relation of smog with the current coronavirus pandemic. Along with negative aspects, positive aspects of the lockdown on air pollution are also discussed in the paper.

## 2. Research Methodology

### 2.1. Identifying the Research Question

The following questions are established to address the relation of smog with coronavirus disease.
What are the components of the smog that are of concern during the pandemic?Does amalgamation of coronavirus and smog increase the health risks?Does the COVID-lockdown bring any positive effects on the air quality?

### 2.2. Finding and Selecting the Relevant Studies

To draft this review, we have searched PubMed for the articles that discussed the relationship between smog and coronavirus disease and have to obtain the most relevant studies using simple keywords “Smog”, “Air pollution”, “Coronavirus” and “COVID-19”. We have also gone through the references section of these articles to select the pertinent publications.

## 3. Smog

The 20th century marks some of the disastrous events related to smog. In the 1930s, the areas of Liège and Huy alongside River Meuse were hubs of industries in Continental Europe. After the industrial revolution fertilizer, glass, zinc smelters, steelworks, and explosive manufacturing plants were established in these areas [25]. At the end of the year, these areas were shrouded by a thick fig for five days (1–5 December). Within 3 days hundreds of people contracted the signs and symptoms of respiratory diseases. The government was baffled completely after the death of 63 people. On the 6th of December, smog disappeared completely with improvement in respiratory troubles [26]. 

On the 28th of November 1939, dwellers of ST. LOUIS faced a thick smog for over a month as they were burning cheap coal to keep themselves warm from cold weather. Kings-highway and neighboring areas were completely covered by darkness during the daytime. That day is attributed as ‘Black Tuesday’ in history. ST. LOUIS faced smog events later in the next year after which the authorities took proper actions to resolve the pollution issue [27,28,29].

On October 26, 1948 fog mixed with industrial pollutants engulfed the atmosphere of Donora, Pennsylvania [30]. Donora Zinc Works, part of US Steel was blamed by the authorities as a major contributor to smog. About 5000–7000 residents became ill, 400 were hospitalized and 20 people died. After five days on 31st October 1948, the smog was dispersed by the rain [31]. Donora also faced small smog events on the 4th and 14th of October, 1923 [32]. After the events, the Donora Zinc Works was shut down [31].

In 1952, London (England’s capital) was engulfed for five days by the lethal black haze called, Great London Smog of 1952 [33]. In December residents of London burned the high sulfur coal [2] to keep themselves warm. This black smoke escaped from their chimneys and mixed with fog [34]. Then this smog cooled by air covered the atmosphere and blocked the sunlight. This black haze proved to be hazardous when converted into sulfur dioxide and sulfuric acid (i.e., corrosive) and affected the eyes, skin, respiratory and cardiac systems of Londoners [33]. This smog caused an increase in hospitalization (48%), respiratory diseases (163%), and asthma in newborn children (20%) [35].

Historical events are discussed to bring an insight into the occurrence of the smog. It helps to determine the possible season, time, or region in which smog is most prevalent. If one knows the possible time of occurrence of smog during the pandemic, then measures can be taken accordingly.

Different types of smog contribute to air pollution. They are London smog (high content of sulfur oxides), Polish smog (PM_10_, PM_2.5_, PM_1_, and various polycyclic aromatic hydrocarbons such as benzo-pyrene), photochemical smog (nitrogen oxides, ozone, hydrocarbons, and VOCs) as shown in Figure 2, and the natural smog released from volcanoes (CO, CO_2_, SO, H_2_, and H_2_S) and plants (hydrocarbons and VOCs) [1,36,37,38,39]. Table 1 shows the distinctive features of different types of smog.

## 4. Some Major Smog Affected Populations

Smog has affected developing as well as under-developing countries likewise. The air quality of any region is estimated by Air Quality Index (AQI). The more the AQI of a region more is pollution in the environment. The AQI values are compared with the units described in Figure 3. There are environment protection agencies that work for the improvement of air quality in the region because it affects nature as well as humans. Therefore, the implication of these standards and regulations during a pandemic is important as they help to monitor the air quality of different regions. These standards make sure that pollutants concentrations do not cross the threshold levels i.e., maximum permitted level (MPL), and if some have already crossed that limit then how could we reduce their concentration to MPLs. Air pollution has become a global problem but we can see that policies regarding control of air pollution vary from region to region. Developed countries like the United States (US) and European Union (EU) have adopted more advanced technologies while developing countries like India and China have just started to build their legislation regarding air pollution. 

### 4.1. China

Air pollution has become a most concerning affair in China. Urbanization is considered the most detrimental cause of air pollution in which rural and agricultural land is converted to urban and non-agricultural land. Moreover, natural habitats are metamorphosed into cities. The enormously increasing Chinese economy, industrialization, and urbanization come at the cost of severe air pollution especially smog pollution [41]. After smoking, high blood pressure, and dietary risks, ambient PM_2.5_, and PM_10_ have become the fourth leading cause of death in China [50]. Nonetheless, the population affected by the recent events of air pollutions in China is phenomenal. Each year 350,000 to 400,000 deaths are attributed to air pollution in China [51]. Beijing faced multiple periods of prolonged air pollution in January 2013. The PM_2.5_ was calculated 32 times higher in Beijing (i.e., 800 mg/m^3^) than that recommended by World Health Organization (WHO) (i.e., 23 mg/m^3^) [52]. Similarly, another episode of smog stuck in Beijing for six days in February 2014. These smog spells affected not only Beijing but also nearby cities forcing the people to stay indoors to prevent adverse health effects [53]. The air quality index (AQI) is the unit used to measure the quality of air in a particular region. The AQI between 0 and 50 is considered good, 50 to 100 is moderate while 101 to onwards is considered unhealthy. Shahecheng (156), Nantong (140), Luancheng (134), Wuda (133), Handan (132), Yangliuqing (127), Dawakou (124), Yigou (122), and Zibo (119) are currently the most polluted cities of China [54]. Being an industrial country China has begun to endorse the policies regarding control of air pollution. Even after the implementation of the Action Plan, 2013 as a strategy to control air pollution, the levels of smog in the atmosphere are still concerning [55]. The evolution of Chinese air pollution control legislation and the standard sets for air pollutants are described in Table 2 and Table 3.

### 4.2. United Kingdom

The Great Smog of London 1952, lessoned the people about the long term health consequences of air pollution. In 1956, Clean Air Act was introduced in England to cope with air pollution [58]. Smokeless burning facilities were announced in heavily polluted cities. Reforestation and the use of eco-friendly fuels are encouraged to reduce air pollution [59]. Despite efforts Ashford (109), Crowborough (108), Faversham (107), Ealing (106), London (104), Shenley (104), Cambridge (103), East Ham (103), Cranbrook (102), and Lewes (102) are most polluted cities of England [60]. United Kingdom followed the European Union laws and standards regarding air pollution as shown in Table 4 and Table 5.

### 4.3. The USA

Everybody has noticed the effects of the horrible brown haze in urban communities like Shanghai, China, or New Delhi, India. However, it is observed that there are issues with air contamination in the USA as well, particularly on the off chance that you live in California, as per the American Lung Association’s 2018. California’s Bay area encounters undeniable degrees of both smog and particulate matter contamination. In the colder time of year, wood smoke from chimneys causes significant degrees of smog [64]. In the USA, North fork (186), Oakhurst (186), Kamiah (184), Orofino (170) Hamilton (164), Moscow (162), McCall (160), La Jolla Shores (158), Lewiston (158), and Pullman (158) are worst cities in context to air pollution [65]. Other than these, Krasnoyarsk-Russia (169), Lima-Peru (163), Kabul-Afghanistan (156), Jakarta-Indonesia (152), Santiago-Chile (152), Tehran-Iran (108), and London-United Kingdom (104) are the top polluted communities of the world according to live stats of Air quality and pollution city ranking of 2021 [66]. The US legislation related to air pollution control have evolved much and has set some standard values for pollutants as shown in Table 6 and Table 7.

## 5. COVID-19 Pandemic

The 2019 novel COVID or the extreme intense respiratory condition COVID-19 (SARS-CoV-2) for what it’s worth presently called, the 2019 novel COVID for what it’s worth presently called, has quickly spread globally [70]. According to the World Health Organization (WHO) and Center for Disease Control (CDC) around 222 M instances of COVID-19 (Coronavirus) and 4.6 M fatalities have occurred till the 8th of September 2021. Most affected populations of the world are the USA (41 M cases), India (33 M cases), Brazil (20.9 M cases), Russia (7 M cases), UK (7 M cases), France (6.8 M cases), Turkey (6.5 M cases), Argentina (5.2 M cases), Iran (5.1 M cases), Colombia (4.9 M cases) and so on 223 countries of the world are affected [71].

Coronavirus is an RNA virus (enveloped) having a diameter of 60 nm to 140 nm. Spike-like projections are present on its surface due to which it has a crown-like structure when observed under the electron microscope that’s why it’s named Coronavirus [72]. Pneumonia-like infection was first observed in Wuhan, Hubei region, China in December 2019 in local workers of the Hunan seafood market. Initially, they faced intense acute respiratory distress syndrome (ARDS) and respiratory failure in critical stages [73]. January 7^,^ 2020, marks the day when SARS-CoV-2 was isolated for the very first time from the throat swabs of the patient. After China, it gradually spread in Thailand, Japan, Korea, and the USA. All first cases reported (26 out of 29) had a travel history to China. The remaining 3 had a meet-up or are relatives of the other 26 patients. This study conducted by WHO members confirmed that the seafood market of China was the epidemiological source of COVID-19 [74]. Moreover, they also concluded that coronavirus spread through human-to-human contact, and no intermediary live host is involved in transmission. Coronavirus infection spreads from symptomatic people through droplets produced from coughing or sneezing as well as asymptomatic people [75]. The disease can also be attained by rubbing your nose, eyes after touching virus-contaminated surfaces. The stool of patients also contains the virus that results in contamination in the water supply [76].

## 6. Smog and Coronavirus

The correlation between air pollution and COVID-19 has pros and cons. Studies have backed both the aspects that COVID-19 mortality rates are high in highly polluted regions while the lockdown during the pandemic may lower the air pollution rates and thus lower the infection rates. Previous studies have proved that smog is a risk factor for respiratory infections by carrying microorganisms to humans and distressing the body’s immunity to make people more vulnerable to pathogens [77,78]. 

Smog effects on our health depend on several different factors, including the level of air pollutants, types of air pollutants, age and health conditions, exposure time, and where you live. Smog affects different organs of the body as shown in Figure 4 and Table 8. It can irritate our eyes, nose, and throat and can also cause existing heart and lung problems in people to worsen or lead to lung cancer if the exposure time of smog is long [3,79]. It also leads to premature death. Studies on ozone have shown that once it gets into your lungs, it can cause damage even when you are feeling well. It affects mainly those people who are at risk or suffer from heart and lung diseases. Children are most sensitive to smog because their respiratory systems are still underdeveloped and they have an active lifestyle [80]. However, these effects vary from person to person and exposure time. Healthy people who are exposed to smog for short period do not get long-term effects but in comparison, if a person is immunocompromised the effects will be long-term and might get worsen if the time of exposure is long and the dose is higher. Children are at more risk than adults, even if a low dose is present [81].

The pathogenesis of SARS-CoV-2 is similar to its closely related SARS-CoV-1 with an exception of the S-protein of SAR-CoV-2 that has a more binding affinity for Angiotensin Converting Enzyme-2 (ACE-2) receptors. Excess of plasma angiotensin-II is accumulated because of down-regulation of ACE2 leading to ARDS and myocarditis making other organs like the esophagus, kidney, lungs, heart, and ileum more vulnerable to SARS-CoV-2 [93]. Children under 5 years have low numbers of ACE2 receptors which probably makes them less susceptible to the disease [94]. 

The symptoms of the coronavirus vary from asymptomatic to severe respiratory diseases and organ damage. Some common symptoms include fever, fatigue, cough, headache, loss of smell and taste. Some people also suffer from acute lung injury (ALI) and impairment in blood clotting. Despite pulmonary damage being the cause of fatality, elderly patients also develop coronary heart diseases, atherosclerosis, ischemic cardiomyopathy, or hypertension. Apart from pulmonary damage COVID-19 is also involved in extra-pulmonary disorders like lymphopenia (67–91% of COVID-19 cases), proteinuria (87%), hepatocellular injuries (14–53%), gastrointestinal damage (12–61%), thrombotic complications (30%) and acute kidney injury (0.5–29%) [95]. The death rate in elder people is more than that of young people [96]. The death rate of adult patients in hospitals ranges from 4–11%, while the overall death rate is considered to range from 2–3%.

## 7. Amalgamation of Smog and COVID-19

Since COVID-19 is a respiratory disease, it is investigated that smog results in the transmission of coronavirus, and SARS-CoV-2 can remain feasible in the air for hours [97]. Short-term exposure to elevated concentrations of air pollutants results in an increased risk of coronavirus infection. The significance of lessening air contamination is perceived under its notable effect on environmental change and its impact on wellbeing because of expanded bleakness and mortality related to smog and air pollution [98]. As per late investigations, smog appears to support the spread of coronavirus disease. As the viral particle is airborne, the impact of COVID-19 is exasperated by smog [99]. Even though there are opposing sentiments on the transmission of SARS-CoV-2, it appears that one can obtain the disease through the air [16] because of its strength in mist concentrates [97] and the reality that the pollutant cloud and its payload (microbe bearing droplets) can travel 7–8 m [100]. In a recent experiment, aerosols containing SARS-CoV-2 were created using three jet Collison nebulizers to mimic the aerosolized atmosphere. The viral load remained active for 3 h even though their virulent capacity was reduced [97]. Similarly, a double hit hypothesis has also been proposed initially in which NO_2_ and PM_2.5_ are considered responsible for coronavirus spread [19]. Particulate matter (PM_2.5_) stabilizes the exhaled droplets in the air after fusing with them. The droplet would have evaporated rapidly in the atmosphere under normal air conditions but in high PM concentration, PM stabilizes the droplet and reduces its diffusion coefficient making it more transmissible. Moreover, a study conducted on mice supported the hypothesis that increased exposure to PM promoted the ACE2 and transmembrane-protease serine2 (TMPRSS2) production in macrophages and angiotensin-receptor type 2 (AT2) in lung tissues. This increase made the mice more susceptible to SARS-CoV-2 [101]. The studies conducted here supported the hypothesis that regions with a high concentration of air pollution are more affected by the coronavirus. Some of these studies are listed in Table 9.

From the above studies, it can be summarized that pollutants especially NO_2_ and PM are strongly responsible for respiratory disorders in humans. Similarly, SARS-CoV-2 is also associated with respiratory disorders. Therefore, the existence of air pollutants and coronavirus at a time can prove to be fatal as described in the earlier studies.

## 8. Impact of Lockdown on Smog

Since the 15th of December 2019, transmission from patients to medical care staff has happened, which shows that human-to-human transmission has occurred through close contact [111]. Most nations have forced city lockdown also, quarantine measures to diminish transmission to manage the epidemic. Public danger correspondence exercises have been performed to improve public attention to self-insurance [112]. The Chinese government has step by step executed a severe lockdown on Wuhan and encompassing urban areas as of 23 January 2020. Not before long, the Government of India also reported a total cross-country lockdown, from the 24th of March 2020. All industries, entertainment centers have been temporarily shut down. Domestic as well as all international flights have been suspended, trains and public transport have been temporarily banned [113].

The lockdowns imposed by governments all around the world have caused economical and financial instability. However, due to lockdowns 30% reduction in air pollutants have been evident in COVID-19 epicenters like Brazil, the USA, Spain, Italy, Wuhan according to reports of the Center for Research on Energy and Clean Air (CREA), European Space Agency (ESA), and National Aeronautics and Space Administration (NASA). NASA and ESA have reported a substantial drop of 2.5µ diameter in (PM_2.5_ and PM_10_) in Beijing, China where most of the pollution comes from heating instruments in the winters and heavy industrialization [24]. During the time of lockdown, air quality and smog conditions would be predicted to have improved in favor of life being. Due to traffic and industrial lockdown, a fall of ~63% in the concentration of NO_2_ is evident in Wuhan, China. This fall in NO_2_ concentration resulted in fewer deaths of people in Wuhan (496 deaths prevented), Hubei (3368 deaths prevented), and in China (10,822 deaths prevented). Similarly, a shortfall of 20 µg/m^3^ in PM_10_ concentration is also observed in Wuhan. However, no reduction was noticed in SO_2_ and CO concentration because of the dependence of the country on coal-based energy plants [114]. 

Another study conducted over Pakistan stated a decrease of 7.39% in PM_2.5_ and 4.13–5.78% drop in column aerosol optical thickness (AOT) [115]. Hernandez-Paniagua and his colleagues concluded that due to the lockdown of motor vehicles, the concentration of NO_2_ and PM_2.5_ decrease significantly in Mexico. However, other pollutants concentration remains almost undisturbed except for an increase in O_3_ concentration [116]. In Ontario, Canada NO and NO_2_ concentration decreased rapidly while O_3_ concentration decreased slowly but PM_2.5_ remained the same [117]. Madrid, Spain faced a downfall in NO_2_ concentration by 62% [118]. In Gujarat, India 30–84% reduction in NO_2_ occurred while O_3_ increased by 16–58% [119]. The atmosphere of Delhi, India got rid of 55% of PM_10_, 49% of PM_2.5_, 60% of NO_2_, and 19% of SO_2_ while Mumbai, India got rid of 44% of PM_10_, 37% of PM_2.5_, 78% of NO_2_ and 39% of SO_2_ [120]. Some figures before and after lockdown are listed in Table 10.

After the lockdown of city traffic, workforce stream control turned into the main perspective. Traffic contamination produces NO, NO_2_, CO, CO_2_, hydrocarbons, and toxins that are injurious to health [121]. There was a distinguishable relationship between traffic-associated air contamination and early mortality, and the danger of respiratory and cardiovascular diseases enlarged in people living close to elevated traffic polluted places [122]. Decreasing the outflows from engine vehicles, particularly trucks and transports, could deliver extensive medical advantages [123]. After lockdown, many surveys were done in hospitals which showed that after a consecutive lockdown of 14 days there was seen a major decline in children in hospitals complaining of asthma problems [124]. In accretion, the decrease in industrial actions after the lockdown also forces definite environmental and health effects. The lockdown has caused financial downfalls in many countries and cities, but it also has given clean air to residents of some of the world’s most contaminated cities. The coronavirus pandemic has led to the decrease in the concentration of pollutants like SO_2_, NO_2_, CO, PM_2.5_ that contribute to smog all over the world and to some extend have enhanced the air quality in most of the polluted cities of the world [125,126].

## 9. Limitations of the Study

This is a narrative literature review that provides a simple insight into the relation between air pollution and coronavirus in large representative populations. The key limitation of this review is that the individual-level risk factors like race, age, and smoking status are not included. Moreover, chances of miscalculation are always there because during the study we assumed that all people in the region are exposed to equal concentrations of air pollution. The relation between smog and COVID-19 is based on area-level studies so the data is useful to develop coping strategies against the situation in a specific area.

## 10. Conclusions

Human activities like the burning of fossil fuels, coal combustion, and the smoke from exhausts of automobiles release toxic gases which react in the atmosphere and give rise to secondary pollutants. All these pollutants collectively contribute to smog. Each year rise in respiratory disease is related to smog episodes. Moreover, cardiovascular diseases, neurological disorders, underdevelopment of fetuses, and cancer are the major diseases that are related to smog pollution. Smog episodes can have deleterious effects amidst the COVID-19 pandemic. When a person is long exposed to air pollution, the coronavirus would have an additive effect on the respiratory and cardiovascular systems of the human. From the studies conducted it seems that particulate matter and nitrogen oxides increase the activity and production of ACE2 which in turn enhances the chances of uptake of SARS-CoV-2 and could damage lungs, heart, and blood vessels. However, the relation between smog and coronavirus isn’t this. They share an ambiguous relation where on the one hand air pollution may worsen the COVID-19 mortality rate, the lockdown imposed because of a pandemic may have some positive aspects as well. During the lockdown periods, a significant decrease in some of the pollutants like NO_2_, SO_2_, and PM have been recorded. For a better future, anthropogenic emissions need to be controlled because vaccines are effective against pandemics and not against air pollution.

## Figures and Tables

**Figure 1 ijerph-18-11408-f001:**
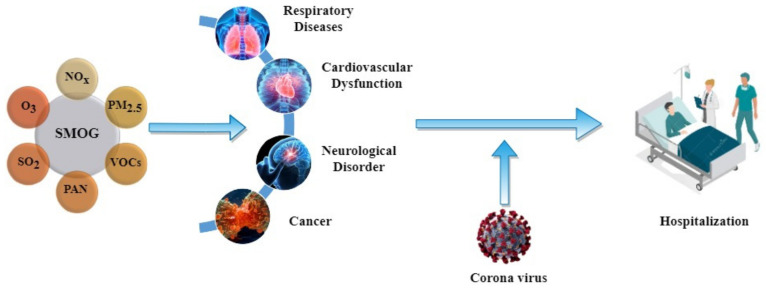
Relationship between COVID-19 and smog.

**Figure 2 ijerph-18-11408-f002:**
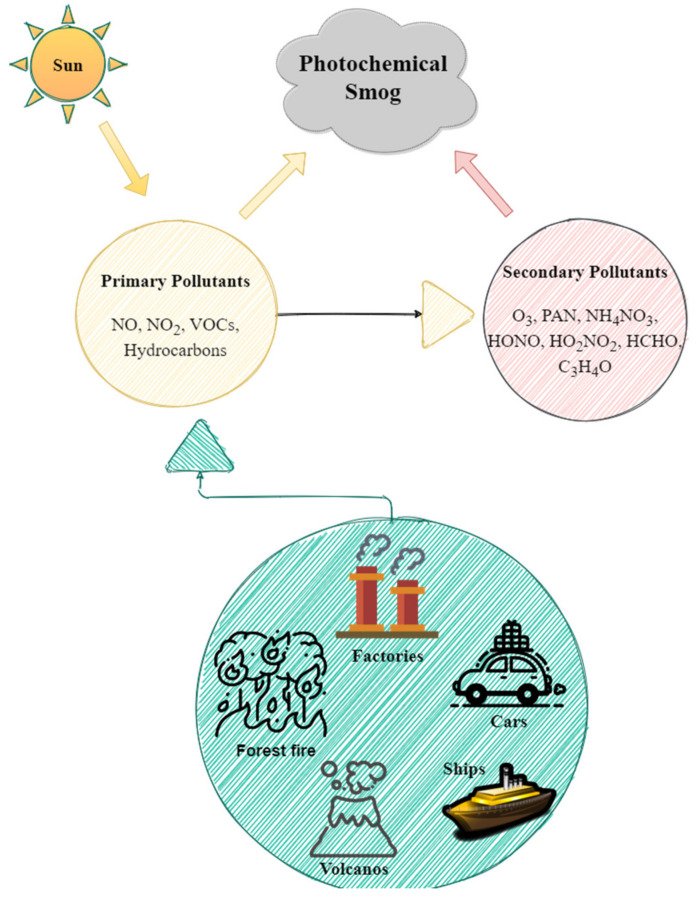
The activities of erupting volcanoes, traffic emissions, forest fires, general combustion, mining, agriculture are directly or indirectly involved in the production of primary pollutants like NO, NO_2_, VOCs, and hydrocarbons which are major forerunners of smog. These primary pollutants undergo chemical reactions in presence of sunlight to form secondary pollutants like formaldehyde, peroxyacetyl nitrate (PAN), and O_3_ [40,41]. Both primary and secondary pollutants then concoct smog.

**Figure 3 ijerph-18-11408-f003:**
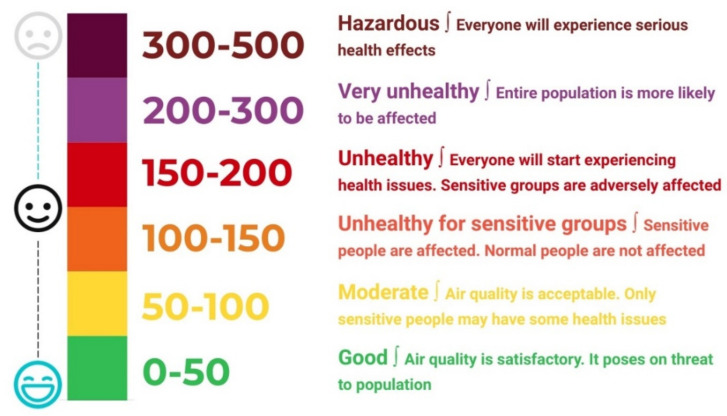
Air Quality Index Chart.

**Figure 4 ijerph-18-11408-f004:**
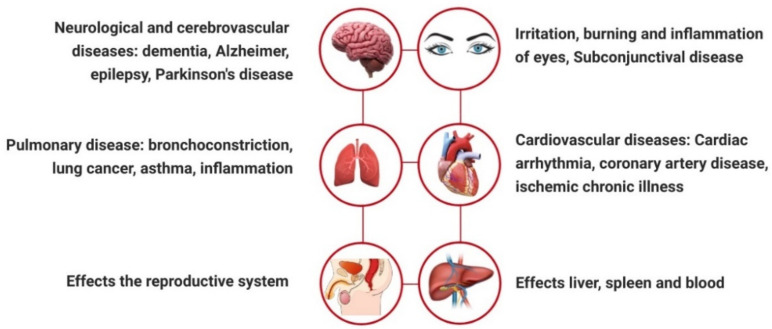
Effect of smog particles on different organs (nervous system, eyes, throat, lungs, heart, liver, spleen, and reproductive system) of the human body and the problems associated with it.

**Table 1 ijerph-18-11408-t001:** Distinctive features between London, Photochemical and Natural smog.

Characters	London Smog (Sulfurous Smog)	Polish Smog	Photochemical Smog (Los Angeles Smog or Summer Smog)	Natural Smog	References
Definition	Develops due to high concentration of sulfur oxides in the air	When the temperature drops, inversion takes place and a low-level cloud of pollutants form a dusty cloud	It is produced when sunlight reacts with oxides of nitrogen or at least one VOC ^1^	It may result due to volcanoes also known as acid smog (vog) and by plants i.e., natural sources of hydrocarbons and volatile organic compounds	[36,42,43,44]
Occurrence	It occurs in cold, humid climates	It occurs in the winter seasons	It occurs in a warm, dry, and sunny climate	It occurs mostly in warm, humid, and summer climate	[36,37,45,46]
Effects	It irritates the eyes, causes bronchitis and lung problems	It affects the lungs, causes asthma and cardiovascular diseases	It irritates the eyes, causes obstructive pulmonary disease, cardiovascular disease, and asthma.	Irritation and inflammation of eyes, dry cough, anterior uveitis, breathing difficulties, asthma, subconjunctival hemorrhage.	[45,47,48,49]

^1^ VOC: Volatile organic compound.

**Table 2 ijerph-18-11408-t002:** Evolution of Chinese legislation for Air pollution control [56].

Year	Law or Action	Description
1979	Environmental Protection Law	First legislation related to environmental pollution was established
1987	Air pollution prevention and control Law	For the control of pollution emissions from industries in specific areas
1989	Environmental Protection Law (EPL)	For the very first time, institutional buildings were constructed for the enforcement of law
1998	Establishing acid rain and sulfur dioxide (SO_2_) control areas	Measures are developed to reduce the acid rain and SO_2_ pollutants in specific areas
2000	Amendment of Air Pollution Prevention and Control Law	Data related to air pollution was linked with AQI ^1^ which was classified as natural, urban, and industrial. Major pollutants were targeted i.e., SO_2_, NO_2_, and PM in 42 cities
2002	Environmental Impact Assessment (EIA) law	“Pollute first, clean up later” model was developed to highlight the sources of pollution
2008	Ministry of Environmental Pollution (MEP)	State administration of Environmental Protection was upgraded to a ministry
2010	ODS ^2^ regulation	Control of ozone by ODS
2013	Air pollution prevention and control action plan	Its purpose was to reduce pollution in specific regions. It aimed to reduce PM by 10% by 2017.
2015	Amendment of EPL	According to these amendments, non-compliance is punished with a high price, EIAs plans should be made mandatory and public awareness programs be done
2016	2nd amendment in Air Pollution Prevention and Control Law	A system for co-operation between regions was introduced. Limits of vehicle emissions were set and involvement of local government was enhanced
2016	Amendment in EIA law	Increases the facilities and planning of EIA
2018	Ministry of Ecology and Environment	The working structure of MEP is enhanced
2018	Environmental Protection Tax law	To replace old pollution fee system
2018	Blue sky war-winning action plan	The second phase of the 2013 plan targets reduction of VOCs, NOx, and ozone in more cities in China

^1^ AQI: Air Quality Index, ^2^ ODS: Ozone-Depleting Substance.

**Table 3 ijerph-18-11408-t003:** National Ambient Air Quality Standards (NAAQSs) of China [57].

Year	No. of Standards	Grade ^1^	CO ^2^	NO_2_ ^3^	SO_2_ ^3^	O_3_ ^4^	TSP ^5^	PM_2.5_ ^6^	PM_10_ ^3^
1982	GB3095–82	I	100	50	50	120	150	-	50
		II	100	100	150	160	300	-	150
		III	200	150	250	200	500	-	250
1996	GB3095–1996	I	100	40	20	120	80	-	40
		II	100	40	60	160	200	-	100
		III	200	80	100	200	300	-	150
2000	Amended GB3095–1996	I	100	40	20	160	80	-	40
		II	100	80	60	200	200	-	100
		III	200	80	100	200	300	-	150
2016	GB3095–2012	I	100	40	20	160	80	15	40
		II	100	40	60	200	200	35	70

^1^ Grade I: Places like forests and national parks, II: Rural, urban, industrial and commercial areas included, III: Heavy industry areas, ^2^ CO: mg/m^3^, 1 h average, ^3^ NO_2_, SO_2_, PM_10_: µg/m^3^, 24 h average, ^4^ O_3_ µg/m^3^, 1 h average, ^5^ TSP: Total Suspended Particle, ^6^ PM_2.5_ µg/m^3^, 1-year average.

**Table 4 ijerph-18-11408-t004:** Evolution of European Union legislation on air pollution [61,62].

Year	Law of Action	Description
1979	Convention on Long-range Transboundary Air Pollution	Focused mainly on pollutants that cause eutrophication and acidification i.e., NOx, cadmium (Cd), lead (Pb), mercury (Hg), NH_3_, PM, SO_2_, and VOCs ^1^.
1980	Directive 80/779/EEC ^2^	This directive dictates the limitation levels of SO_2_ and PM
1982	Directive 82/884/EEC	Lead limitations were set
1985	Directive 85/203/EEC	This directive highlighted the NO_2_ limitation levels. It did not apply to the inside buildings
1988	Directive 88/609/EEC	The purpose of this Council directive was to limit the emissions of certain pollutants i.e., NOx and SO_2_ for large combustion plants. It helped in the reduction of these pollutants.
1992	Directive 92/72/EEC	It introduced provisions related to tropospheric O_3_. It holds the Environment Protection Agency (EPA) responsible for measuring O_3_ concentrations and defining the threshold levels of O_3_ for industries.
1996	Directive 96/61/EC	It’s a directive on Ambient Air Quality Assessment and Management which aims at preventing the harmful effects of pollutants on the environment as well as human health. As this directive failed four daughter directives with more specificity were introduced
1999	1st daughter directive 1999/30/EC	It focused on the limit levels of NO, NO_2_, SO_2_, lead (Pb), and dust. It aimed to protect the ecosystem, plants, and humans.
2000	2nd daughter directive 2000/69/EC	It defined the benzene and CO acceptable levels. Aims at the protection of humans
2002	3rd daughter directive 2002/3/EC	Ozone levels in ambient air. Aims at the long-term protection of plants and humans.
2004	4th daughter directive 2004/107/EC	It defined the acceptable levels of nickel, cadmium, arsenic, and PAHs ^3^. Aims to protect humans.
2008	Directive 2008/50/EC	It’s a directive on ambient air quality and cleaner air for Europe. It replaced the directives from 1996 to 2002. It ensures the enforcement of laws regarding air pollution. It urges regional authorities to take measures according to their environmental conditions. If any region surpasses the threshold level of pollutants, this directive provides a deadline for reducing the pollutant levels to the threshold.

^1^ VOCs: Volatile organic compounds, ^2^ EEC: European Environment Agency, ^3^ PAHs: Polycyclic aromatic hydrocarbons.

**Table 5 ijerph-18-11408-t005:** Ambient Air Quality Standards of European Union [63].

Pollutants	Average Time	Concentration	Exceed Permitted Each Year
Carbon monoxide (CO)	8 h	10 mg/m^3^	-
Ozone (O_3_)	8 h	120 µg/m^3^	Average of 25 days in 3 years
Sulfur dioxide (SO_2_)	1 h	350 µg/m^3^	24
	24 h	125 µg/m^3^	3
Lead (Pb)	1 year	0.5 µg/m^3^	-
Nitrogen dioxide (NO_2_)	1 h	200 µg/m^3^	18
	1 year	40 µg/m^3^	-
PM_2.5_	1 year	25 µg/m^3^	-
PM_10_	24 h	50 µg/m^3^	35
	1 year	40 µg/m^3^	-
Arsenic (As)	1 year	6 ng/m^3^	-
Benzene	1 year	5 µg/m^3^	-
Nickel (Ni)	1 year	20 ng/m^3^	-

**Table 6 ijerph-18-11408-t006:** Evolution of U.S. legislation on air pollution [67,68].

Year	Law or Action	Description
1955	Air Pollution Control Act	In 1948, a 5 days event of smog in Donora, an industrial town, in Pennsylvania prompted the passing of the first air quality act in the U.S. In 1955, air pollution was declared a national problem under Air Pollution Control Act and research on air pollution was funded.
1963	Clean Air Act (CAA) sets Nationwide Air Quality Standards	Under this act, public education programs were carried out and researches regarding control of air pollution were supported. However, it has no intention of reducing the air pollutants
1965	Motor Vehicle Air Pollution Control Act	With some amendments in CAA, standards regarding automobile emissions were laid down.
1967	Air Quality Act (AQA)	This act distributed the responsibilities to the regions to develop and implement control measures against air pollution. However, this wasn’t effective
1970	Clean Air Act Amendments of 1970	A new CAA was passed to control six pollutants i.e., CO_2_, NO_2_, CO, O_3_, PM, and lead. It also provided flexibility to Motor Vehicle Air Pollution Control Act. EPA ^1^ was established to make sure the implementation of the act.
1977	Clean Air Act Amendments of 1977	It is concerned with provisions for the Prevention of Significant Deterioration (PSD) of air quality in areas fulfilling NAAQS ^2^ as well as areas not attaining NAAQS.
1990	Clean Air Act Amendments of 1990	These revisions expanded the limits and responsibilities of the federal government. New amendments were made regarding control of acid rains, air toxins, O_3_ depletion, and ground levels of O_3_. EPA was authorized more responsibilities to enforce air control acts and reduce air pollutants

^1^ EPA: Environment Protection Agency, ^2^ NAAQS: National Ambient Air Quality Standards.

**Table 7 ijerph-18-11408-t007:** National Ambient Air Quality Standards (NAAQSs) of United States [69].

Pollutants	Average Times	Primary Standards ^1^	Secondary Standards ^2^	Exceed Permitted
Carbon monoxide (CO)	1 h	8 ppm	-	<1 per year
	8 h	35 ppm	-	
Ozone (O_3_)	8 h	0.070 ppm	0.070 ppm	4th highest average max 8 h concentration, averaged over 3 years
Sulfur dioxide (SO_2_)	1 h	75 ppb	-	99th% of max 8 h concentration, averaged over 3 years
	3 h	-	0.5 ppm	<1 per year
Lead (Pb)	3 months	0.15 µg/m^3^	0.15 µg/m^3^	-
Nitrogen dioxide (NO_2_)	1 h	100 ppb	-	98th% of max 8 h concentration, averaged over 3 years
	1 year	53 ppb	53 ppb	Annual mean
PM_2.5_	24 h	35 µg/m^3^	35 µg/m^3^	Annual mean
	1 year	12 µg/m^3^	15 µg/m^3^	Annual mean
PM_10_	24 h	150 µg/m^3^	150 µg/m^3^	<1 per year

^1^ Primary standard: Covers human health and sensitive groups (asthma patients and children), ^2^ Secondary standards: Protects human welfare (plants, buildings, and animals).

**Table 8 ijerph-18-11408-t008:** Sources and diseases associated with sulfur dioxide, hydrocarbons, peroxyacetyl nitrate, nitrogen oxide, tropospheric ozone, and particulate matter i.e., different components of smog.

Smog Components	Source	Effect	References
Sulfur dioxide	Industries, burning of fossil fuels, electric generation plant, volcanic eruption	Respiratory problems i.e., irritation, inflammation, and infection. Asthma and reduced lung function. Chronic obstructive pulmonary disease. Cardiovascular disease, cardiac arrhythmias, hemorrhagic stroke	[82,83,84]
Hydrocarbons	Automobile exhaust and industries	Carcinogenic, may cause leukemia, lung cancer	[85]
PAN ^1^	Photochemical reaction of hydrocarbons and nitrogen oxide	Irritation in the eye. nose and throat, breathing problems, damage to proteins	[86]
Nitrogen oxide	Combustion of fossil fuels, volcanic action, lightning, forest fires	Effects liver, spleen, and blood, kidney cancer, prostate cancer, brain cancer, reduce the birth length	[87]
Tropospheric ozone	Formed as a by-product of photochemical smog	Eye and respiratory irritation, cardiovascular disease, heart failure, breast cancer, fatal bladder cancer. Effects growth and bodyweight of the baby throughout pregnancy.	[88,89,90]
PM ^2^	Vehicles, industries	Particles penetrate deep into the lungs, affect the reproductive system, cause Parkinson’s disease, low birth weight, and halt fetal growth.	[91,92]

^1^ PAN: peroxyacetyl nitrate, ^2^ PM: Particulate matter.

**Table 9 ijerph-18-11408-t009:** Studies correlating high COVID-19 incidences to the high rates of air pollution.

Region	Study	References
England	This study suggested that people who have been exposed to chronic levels of air pollution may have a high instance of contracting severe COVID-19. This may be attributed to the weakening of immune defense protocol by air pollution. It has also been suggested that mortality of COVID-19 may also be associated with cytokine storm syndrome, a response of the immune system that ascends to the chain of destructive events in the body and eventually causes death.	[102]
France	A correlation between air pollution and COVID-19 hospitalization maps has been studied. It was evident that areas with high requirements of hospitalization due to COVID-19 have also profound levels of PM_2.5_.	[103]
Czech Republic	In industrialized regions, high air pollution trends correlate with COVID-19 hospitality.	[103]
Poland	Mazowieckie Voivodship, Upper Silesian Voivodship, and Lower Silesian Voivodship hold a maximum number of COVID-19 cases. All these regions have PM_2.5_ concentrations in the range 19.58–29.84 µg/m^3^ which is higher than those set by WHO i.e., 25µg/m^3^.	[104]
United States	Just 1 µg/m^3^ increase in PM_2.5_ concentration causes a 15% increase in COVID-19 fatality rates.	[105]
United States	The increase of 4.6 ppb in a concentration of NO_2_ caused an increase in the mortality rate of COVID-19 up to 16.2%. If this 4.6 ppb concentration could been reduced it would have prevented 14,000 deaths of COVID-19 patients.	[106]
United Kingdom	Out of the first 44,000 deaths of COVID-19, 6,100 (14%) deaths could be attributed to air pollution.	[107]
Germany	Long term exposure to air pollution is involved in 26% of COVID-19 fatalities.	[108]
Lima	A higher concentration of PM_2.5_ is responsible for the increase in COVID-19 cases however it does not affect the rate of COVID-19 fatalities.	[109]
Italy	Most COVID-19 affected regions had a high concentration of PM_2.5_ and PM_10_ during February 2020.	[110]

**Table 10 ijerph-18-11408-t010:** Relative percentage difference of pollutants before and during the lockdown in different regions of the world [115].

Pollutants	Region	Before Lockdown	During Lockdown	Relative Percentage Difference (%)
SO_2_ (µg/m^3^)	Hubei (China)	15.81	13.83	−13.36
SO_2_ (DU) ^1^	Malaysia	1.42	0.99	−35.68
	Sale (Morocco)	6.6	3.3	−0.49
CO (ppbv) ^2^	Chennai (India)	44.1	45.2	−2.46
	Delhi (India)	1.03	0.72	−30.35
	Hubei (China)	1.207	1.02	−16.79
	Kolkata (India)	0.6	0.5	−18.18
	Malaysia	0.8	0.49	−48.06
PM_2.5_	Chennai (India)	29.38	27.33	−7.23
	Delhi (India})	80.51	37.75	−53.11
	Hubei (China)	81.83	65.81	−21.70
	Malaysia	32.3	22.34	−36.46
	Sao Paulo (Brazil)	12.9	12.5	−3.6
	Wuhan (China)	65.5	40.11	−48.08

^1^ DU: Dosbin Unit, ppbv: ^2^ Parts per billion by volume

## Data Availability

No new data were created or analyzed in this study. Data sharing does not apply to this article.

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
