# Peer review of "The Potential Impact of Smog Spell on Humans’ Health Amid COVID-19 Rages"

_ijerph, 2021, doi:10.3390/ijerph182111408_

Round 1
Reviewer 1 Report
In my opinion, the reader would expect more from this manuscript based on the title and not a general overview of the smog. It is clear that the authors have done a lot of work on this review, as it is also shown by the collected references.
The manuscript can be divided into two main parts, which are as follows:
- General overview of the smog: historical events, types of smog, air pollution in different countries, air quality standards and regulations, health effects on human health (approx. 15 pages of the document)
- The relationship between smog and COVID-19: COVID-19 basics, smog and corona, impact of lockdown on smog (approx. 4 pages of the document)
Some comments for the general overview of the smog are listed below:
Lines 106-113: Please check the type of chemical reactions and the names of the compounds. Was the SO2 oxidation via the hydroxyl radical described?
Line 122: benzo(a)pyrene
Line 240: Does the number 17 mean the year 2017?
4. Smog affected populations: There are many AQI values used in different regions or countries. Is the Chinese AQI index shown in figure 4? What do the ranges mean? Are these the percentage of the limit values?
It was mentioned that the AQI values were compared with the units described in figure 4. However, only the Chinese concentration data were compared. Air quality data are not mentioned for Pakistan, United Kingdom and USA in this subchapter.
The figures should be illustrated in a more scientific form.
Some comments for the relationship between smog and corona:
It is not preferred that this part consists of only 4 pages if this review with the title of the potential impact of smog spell on humans’ health amid 2 corona rages. However, the topic of this review is interesting and current. This review should be focused on new findings on air pollution and COVID-19 pandemic in more detailed.
Author Response
The Editor
International Journal of Environmental Research and Public Health
Subject: Submission of revised manuscript ID: ijerph-1333129
Dear Sir
It is stated that I want to submit revised article entitled, “The potential impact of smog spell on humans’ health amid corona rages”. We are highly thankful to referees whose comments helped in improving this manuscript. We have revised the entire manuscript for language as well as for proper flow of the information. Mostly these are the reviewers’ observations which are addressed in the point by point rebuttal file and also incorporated the same in the text. Below is response to referee comments:
REVIEWER 1
Review comment: Introduction is very hard to follow the main focus of the study. Information is too redundant and unclear why the authors mention two broad aspects in their study. As this special issue is related to current pandemic, the authors should focus on the second aspect of their study i.e., smog and current pandemic. Consider sticking to the issues of air pollution and COVID-19 in more details. All information related to the first aspect of the study (lethal effects of smog and air pollution on health) should be removed.
Author’s response: We are very thankful to the reviewer for pointing out the mistakes. Introduction has been modified as suggested. The data regarding Lethal effects of the smog on human health has been compressed to the minimum. More detail have been added to the second aspect pf the study.
Review comment: Provide general overview of the smog (historical events, types of smog, air pollution in different countries, air quality standards and regulations) succinctly.
Author’s response: We are thankful to the worthy reviewer for pointing out an important flaw. In the revised manuscript all these heading are compressed to minimum.
Review comment: The research strategy is not included in the original manuscript. I suggest adding methodological issue after introduction. This section should include the study design more suitable to your research questions. The inclusion/exclusion criteria for studies regarding each topic of interest should be included. The type of studies selected should reflect your research question.
Author’s response: We have added the research methodology and the research questions after the introduction in the revised manuscript.
Reviewer comment: Authors may want to provide the summary of recent studies based on their main topics in the table. For each topic the number of studies selected and a brief description of them should be reported (i.e., the number of time series, cross-sectional, or cohort studies, the number of studies from the US, Europe, other).
Author’s response: We are thankful to the worthy reviewer for thoroughly studying the article. It is a narrative literature review. Therefore, the Preferred Reporting Items for Systematic Reviews and Meta-Analyses (PRISMA) guidelines are not followed here as they are followed in the systematic reviews and meta-analysis. The tables regarding studies are included separately in each of the heading.
Review comment: Discuss the possible limitation of the present review.
Author’s response: This suggestion of the reviewer is noteworthy. Limitations of this paper are added in the revised paper
Thank you once again for your valuable comments. I am available if there are any further queries.
Best Regards
Reviewer 2 Report
I think it is a good narrative review article that describes the health effects of smog, the effect of smog on covid-19, the effect of lock down, etc. in detail and broadly overall. However, because the length of the article is too long, I think that the description in the first half other than the effects of covid-19 and lockdown needs to be summarized somewhat. These are already mentioned in other papers, but if the authors decide they are all necessary, you can leave them as they are. Once again, congratulations on writing a good article.
Author Response

(The authors gave the same response as above.)

Round 2
Reviewer 1 Report
The revised version has been improved.
Author Response
Thank you for your comments
This manuscript is a resubmission of an earlier submission. The following is a list of the peer review reports and author responses from that submission.
Round 1
Reviewer 1 Report
Dear authors,
please, read carefully the pdf attached to this message.
Best.

Reviewer 2 Report
It is an easy to read paper yet there are two major concerns that should be addressed by the authors:
- There is a complete lack of literature review, as well as of the evolution of related regulation on air emissions, in the text. An analysis per 'industry' or 'activity' or air emission regulations should be included, in a similar way to -indicatively- Schinas, O. (2020) Greening the terminals: Operational and policy considerations , doi:10.1007/978-3-030-39990-0 10. It is important for the reader to understand also the limits set by regulators.
- The novelty / contribution to new knowledge of this paper is not adequately presented or conveyed to the reader. Much of the information presented is interesting, as it is summarized, 'condensed' input from other works, yet Sections 5, 6, and 7 need some 'beefing up'.
- The text needs some improvement in expressions in English, see e.g. 'pregnant ladies' in abstract vs 'pregnant women', etc.